# Mapping Irrigated Croplands from Sentinel-2 Images Using Deep Convolutional Neural Networks

**Wei Li** [1,2,3], **Ying Sun** [4], **Yanqing Zhou** [1,2,3], **Lu Gong** [1,5], **Yaoming Li** [2,3,6] **and Qinchuan Xin** [2,4,6,*]

[1] College of Ecology and Environment, Xinjiang University, Urumqi 830046, China
[2] Xinjiang Institute of Ecology and Geography, Chinese Academy of Sciences, Urumqi 830011, China; lym@ms.xjb.ac.cn
[3] University of Chinese Academy of Sciences, Beijing 100049, China
[4] School of Geography and Planning, Sun Yat-sen University, Guangzhou 510275, China
[5] Key Laboratory of Oasis Ecology of Education Ministry, Urumqi 830046, China
[6] CAS Research Center for Ecology and Environment of Central Asia, Urumqi 830011, China
[*] Correspondence: xinqinchuan@ms.xjb.ac.cn

**Abstract:** Understanding the spatial distribution of irrigated croplands is crucial for food security and water use. To map land cover classes with high-spatial-resolution images, it is necessary to analyze the semantic information of target objects in addition to the spectral or spatial–spectral information of local pixels. Deep convolutional neural networks (DCNNs) can characterize the semantic features of objects adaptively. This study uses DCNNs to extract irrigated croplands from Sentinel-2 images in the states of Washington and California in the United States. We integrated the DCNNs of 101 layers, discarded pooling layers, and employed dilation convolution to preserve location information; these are models which were used based on fully convolutional network (FCN) architectures. The findings indicated that irrigated croplands may be effectively detected at various phases of crop growth in the fields. A quantitative analysis of the trained models revealed that the three models in the two states had the lowest values of Intersection over Union (IoU) and Kappa, i.e., 0.88 and 0.91, respectively. The deep models' temporal portability across different years was acceptable. The lowest values of recall and OA (overall accuracy) from 2018 to 2021 were 0.91 and 0.87, respectively. In Washington, the lowest OA value from 10 to 300 m resolution was 0.76. This study demonstrates the potential of FCNs + DCNNs approaches for mapping irrigated croplands across large regions, providing a solution for irrigation mapping. The spatial resolution portability of deep models could be improved further by designing model architectures.

**Keywords:** irrigated croplands; land cover mapping; convolutional neural networks; Sentinel-2 image

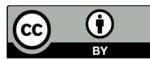

## 1. Introduction

Irrigation refers to the artificial supply of water to the land or soil to nourish crops and increase crop yields, while rainfed crops rely solely on precipitation for their water supply [1]. Irrigated croplands provide nearly 40% of global food yields, and account for approximately 20% of worldwide croplands and 70% of freshwater withdrawals [1]. Over the past half century, irrigated farmland has escalated by 70%, resulting in more than doubled global water consumption due to irrigation. The rise in water usage associated with irrigation has the potential to impact nearly 4 billion individuals worldwide [2]. As such, it is necessary to map and quantify areas of irrigation in order to improve water resource management and allocation to reduce the impacts of global climate change and urban expansion on cropland [3].

Remote sensing has become a powerful technology for mapping land use and land cover across large areas, including monitoring irrigated land use. Some of the existing

land cover mapping products derived from satellite data have included irrigation as an individual class of land cover. For instance, land cover products such as International Geosphere-Biosphere Programme Data and Information System Cover (IGBP DISCover) [4], Global Land Cover 2000 (GLC2000) [5], European Space Agency (ESA) Glob Cover [6], European Space Agency Climate Change Initiative Land Cover (ESA-CCI-LC) [7], and Global Land Cover with Fine Classification System at 30 m (GLC_FCS30) [8] provide land cover maps that delineate irrigated areas. Remote sensing data have also been utilized to create thematic maps of irrigated farmlands. The published products consist of the global map of irrigation areas (GMIA) [9]; global irrigated area map (GIAM) [10]; global rain-fed, irrigated, and paddy croplands (GRIPC) [11]; Global Cropland Area Database (GCAD) [12]; Global Food Security-support Analysis Data (GFSAD) [13]; MODIS-based Irrigated Agriculture Dataset for the U.S. (MIrAD-US) [14]; and irrigated areas at 500 m in China [15]. Most of the aforementioned satellite-based products have a spatial resolution of 250 m or coarser, with both GLC_FCS30 and GFSAD providing maps at 30 m resolution. Depending on spatial complexity and crop field sizes, localized applications often require irrigation maps with significantly higher spatial resolution [16].

Using remote sensing data with a higher spatial resolution should lead to more accurate identification of irrigated cropland, especially in regions with fragmented cropping patterns, such as the Central Valley in California, USA, which is known for its diverse agricultural production. Some research endeavors have aimed to map irrigated areas using Landsat data with a resolution of 30 m. Xie et al. [17] gathered training samples through the maximum enhanced vegetation and greenness index derived from Landsat, as well as the MIrAD-US product, to delineate irrigated croplands throughout the conterminous United States (CONUS) at a 30 m resolution. Moreover, Xie and Lark [18] refined the process of collecting training data by estimating the ideal thresholds for crop greenness and gathering data on the status of center-pivot-irrigated and non-irrigated fields. Ren et al. [19] mapped irrigated and non-irrigated corn for Nebraska at a 30 m resolution by employing training data generated from the MODIS-derived irrigated and non-irrigated map and images from Landsat data. Magidi et al. [20] mapped irrigated areas using Landsat and Sentinel-2 images as well as Google Earth Engine. Yao et al. [21] proposed the mapping of irrigation by soil water content, which is determined using the optical trapezoid method based on various Landsat data. Furthermore, some studies have explored mapping irrigated areas at a spatial resolution scale of 10 m or higher [22–25].

The public release of Sentinel images makes it possible to perform large-scale land cover mapping at a 10 m resolution. Bazzi et al. [22,26–33] proposed detecting irrigation using both SAR-based and optical-based metrics. Specifically, they obtained the time series of Sentinel-1 backscattering coefficients in vertical–vertical and vertical–horizontal polarizations. Subsequently, they applied the PCA (principal component analysis) and the WT (wavelet transformation) to the SAR temporal series to obtain SAR-based metrics. They also selected samples of irrigated and non-irrigated plots based on the SAR-based metric and optical-based metric derived from the NDVI (normalized difference vegetation index) time series of the Sentinel-2 data.

Mapping irrigated areas from remote sensing images involves the task of land cover classification, which aims to assign a predefined land cover class to each pixel in the images. Many studies adopted shallow classifiers, such as maximum likelihood classification (MLC) [34], random forest (RF) [35], support vector machine (SVM) [36], and multi-layer perceptron (MLP) [37], for land cover classification. These shallow classifiers commonly rely on manually-engineered features to perform the feature representation and classification tasks for remote sensing images. As the spatial resolution of remote sensing images increases, the complexity and richness of object details and semantic content also increase, imposing higher requirements on image processing. Shallow classifiers exhibit limited discriminative power in terms of accurately classifying high-resolution images due to the insufficient expression of complex ground information in manually-engineered features [38,39]. To effectively map land cover classes from high-spatial-resolution remote sensing

images, it is crucial to consider the semantic information of objects in addition to the spectral and spatial information of local pixels [40]. Deep convolutional neural networks (DCNNs), a type of deep multi-layer learning model including convolutional layers with multiple layers, have demonstrated promising results for land cover classification using remote sensing data [41]. These models can extract low-level and high-level features from data and achieve robust classification results regarding remote sensing imagery [42]. Compared to shallow classifiers, the deep network model has the advantage of learning and characterizing objects' semantic, spectral, and spatial features [43].

A customary convolutional neural network (CNN) model generally comprises a succession of layers with distinct functions, and its structure mainly includes three types: a convolutional layer, a pooling layer, and a fully connected layer [44]. As a rule, classification accuracy is greater when the CNN layers are deeper [45–48]. The fully convolutional networks (FCNs) can assign dense category labels to images and retain a fine spatial information structure without post-segmentation processing [49]. Several studies on image semantic segmentation models have been implemented by combining DCNN and FCN architecture. For instance, U-Net [50] comprises two parts: down-sampling performed by DCNNs to extract features and up-sampling to restore the feature map size and generate segmentation images. Between down-sampling and up-sampling, skip connections are employed to concatenate underlying and deep information in order to merge features. FCNs have been applied to ground object recognition and classification tasks for remote sensing images [51–55].

There have been instances of utilizing FCNs for high-resolution irrigation extraction [52,53], but some of these studies have only extracted the irrigated fields served by center pivot, which are relatively easy to distinguish from other land cover classes. Deep learning largely relies on training data, making it particularly essential to acquire large and accurate samples for deep learning classification [54]. To extract target objects on a large scale from remote sensing images with a 10 m resolution, many accurate samples are required, necessitating extensive field investigation, screening, and labeling. Arable land is often intermixed with other land types with complex and diverse topography, resulting in small fragmented landscape patches. Additionally, distinguishing between irrigated and rainfed farmland and other vegetation classes on remote sensing images is more challenging during high-precipitation or wetter conditions [22].

This study aims to explore the FCN architecture models, which incorporate DCNN layers for feature extraction which allow us to map irrigated areas from Sentinel-2 images. CNN can automatically learn representative and distinguishing features from raw remote sensing imagery hierarchically, providing powerful descriptions and generalizations for solving large-scale classification problems [55]. The models adopted a ResNet-101 of 101 layers as an encoder block and bilinear interpolation operation as a decoder block. To ensure accuracy, we used the spatial data on irrigation [56,57] released by official agencies to generate samples. We evaluated the models' extraction performances on irrigated areas with varying distribution patterns and irrigation types. Lastly, we analyzed the temporal portability of the deep models in different years as well as the spatial resolution portability at different resolutions of the deep models.

## 2. Materials and Process

### 2.1. Study Areas

Both Washington and California were chosen as the study sites. Washington and California have different climates, topography characteristics, and agricultural systems containing different crop types. Figure 1 shows the geographical location and irrigation distribution within the study regions. The irrigation maps of Washington and California correspond to 2020 and 2019, respectively [56,57].

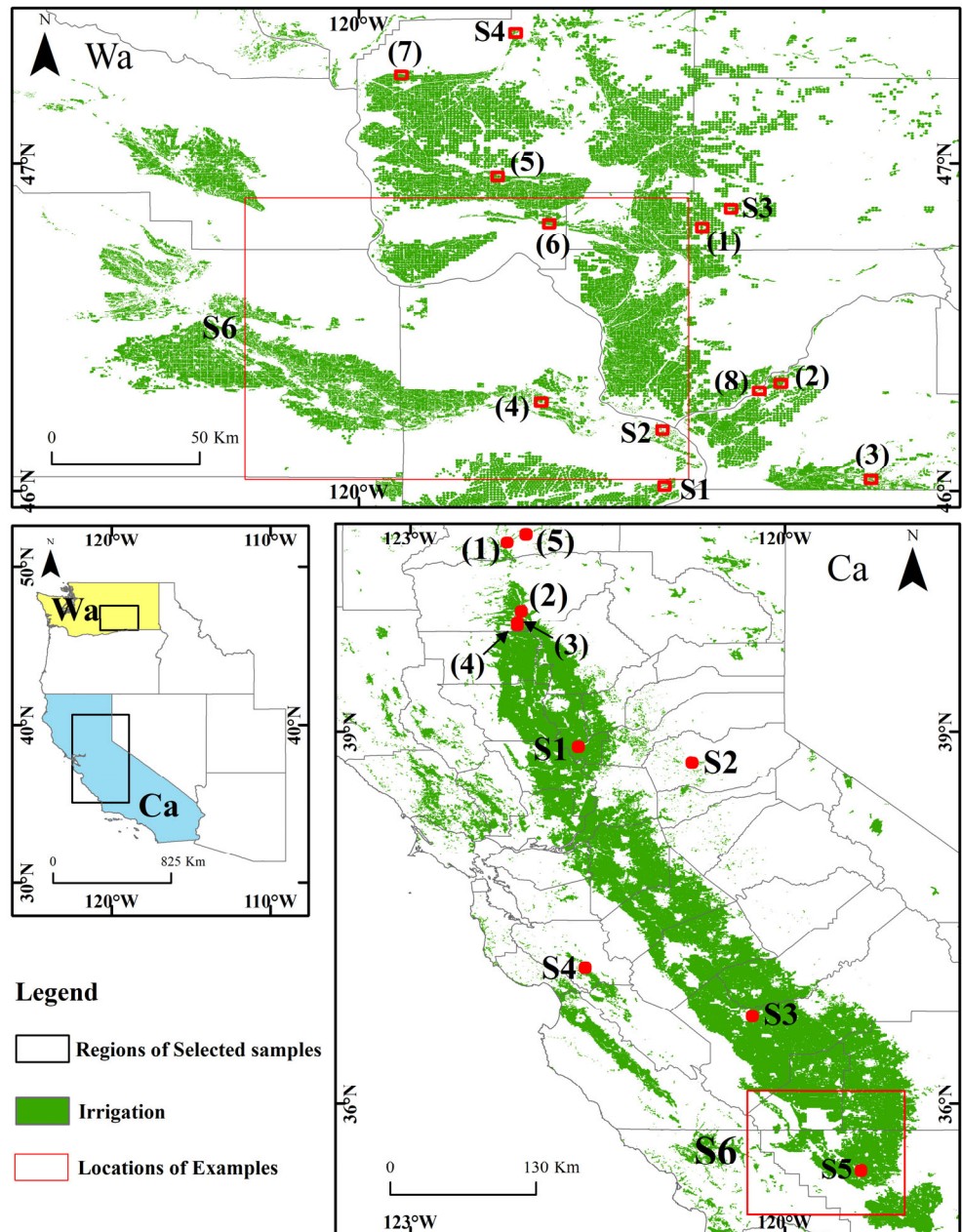

**Figure 1.** Irrigation maps overlaid with state boundaries for the two study areas. The marked subsets display the locations of the selected examples of irrigation extraction results: S1–S4 are the locations of the examples with different irrigation distributions in the two areas; (1)–(8) are the locations of the examples for 8 irrigation categories in Washington; (1)–(5) are the locations of the examples for 5 irrigation categories in California; S5 is the location of the examples of the extraction results in different years in California; and S6 is the location of the examples of the extraction results at multiple resolutions from 10 to 300 m in the two areas.

Washington is located in the northwestern part of the continental United States, with $7.44 \times 10^6$ acres of general farmland and $1.87 \times 10^6$ acres of irrigated farmland [58]. Washington is divided into two parts by the Cascade Mountain Range: the west of Washington and the east of Washington. In the west of Washington, there is a mild climate with abundant precipitation and many rivers, lakes, mountains, and hills. There is a climate with little annual precipitation and plenty of sunshine in the east of Washington. The central area east of Washington consists of plains, known as the Columbia Basin, where the rich volcanic soil and dry weather make it suitable for farming. In the Columbia Basin, dryland

crops include barley, dry peas, lentils, and hay. Irrigated crops include winter wheat, potatoes, vegetables, fruits, hops, and mint.

California has an area of $1.67 \times 10^7$ acres of general farmland and $8.41 \times 10^6$ acres of irrigated farmland [58]. There are dramatically varying and contrasting climates and landscapes in California, including rainy northern coasts, the arid Colorado Desert in the south, Mediterranean-style central and southern coasts, and volcanic plateaus in the northeast. Farm produce in California comes primarily from irrigated farmland, i.e., mainly cattle, milk, cotton, and grapes. About half of the agricultural products in California come from the Central Valley, where irrigation facilities serve crops well. Many of California's large farms are highly intensive, and most of them specialize in one or two crops.

### 2.2. Collection and Pre-Processing of Remote Sensing Data

We used Sentinel-2 level 2A (L2A) optical images from the sentinel-2 mission (SENTINEL-2), which were acquired through Google Earth Engine (GEE). The SENTINEL-2 comprises twin satellites (Sentinel-2A and Sentinel-2B), both with an altitude of 786 km. SENTINEL-2 is equipped with an optical multispectral sensor that samples 13 spectral bands from visible and near-infrared to short-wave infrared at different ground spatial resolutions of 10, 20, and 60 m. The L2A represents the processing level of Sentinel-2 images and implies that the images have already been geometrically rectified and calibrated for atmospheric correction, thus containing bottom-of-atmosphere reflectance values suitable for assessing crop growth. Each granule, called a tile, is a $100 \times 100$ km$^2$ ortho-image in UTM/WGS84 (Universal Transverse Mercator/World Geodetic Syste-1984 Coordinate System) projection. The bands used for analysis were red, green, blue, and near-infrared response (NIR).

To standardize the data across years, we collected temporal medians of images, which were captured monthly. All available sentinel-2 images starting on February 1 and extending until October 31 (9 monthly periods) were collected and sorted by time using GEE. For each nonoverlapping monthly segment within the nine months, the temporal median of all images acquired during that segment was computed, and the resulting median image was stored and used as the feature of the model. The final features fed into the model were nine temporal median images (each comprised of four bands, red, green, blue, and NIR, for a total of thirty-six bands) sorted by time. We treated each band as a separate feature without explicit temporal information. The detailed basis for the data treatment can be found in [59].

In Washington, we collected the remote sensing images acquired in 2020 to train the models. In California, we collected the remote sensing images acquired in 2019 to train the models and the images acquired in 2018, 2020, and 2021 to discuss the extraction effect of the deep models in other years. The extraction effect of deep models trained on images with resolutions of 10 m and on images with different spatial resolutions was examined in Washington and California. Images with resolutions of 90, 100, 200, and 300 m were generated by applying resampling operations to existing Sentinel-2 L2A images with resolutions of 60 m. All samples, including the images and corresponding labels, were cropped into images with sizes of $256 \times 256$ pixels.

### 2.3. Irrigated Cropland Layer Dataset

In order to extract irrigation from high-resolution remote sensing images, the deep learning algorithm usually requires many accurately labeled samples as training data. Figure 2 illustrates typical examples of irrigation and non-irrigation for the two study areas. Details of the irrigated farmland layer datasets are explained below.

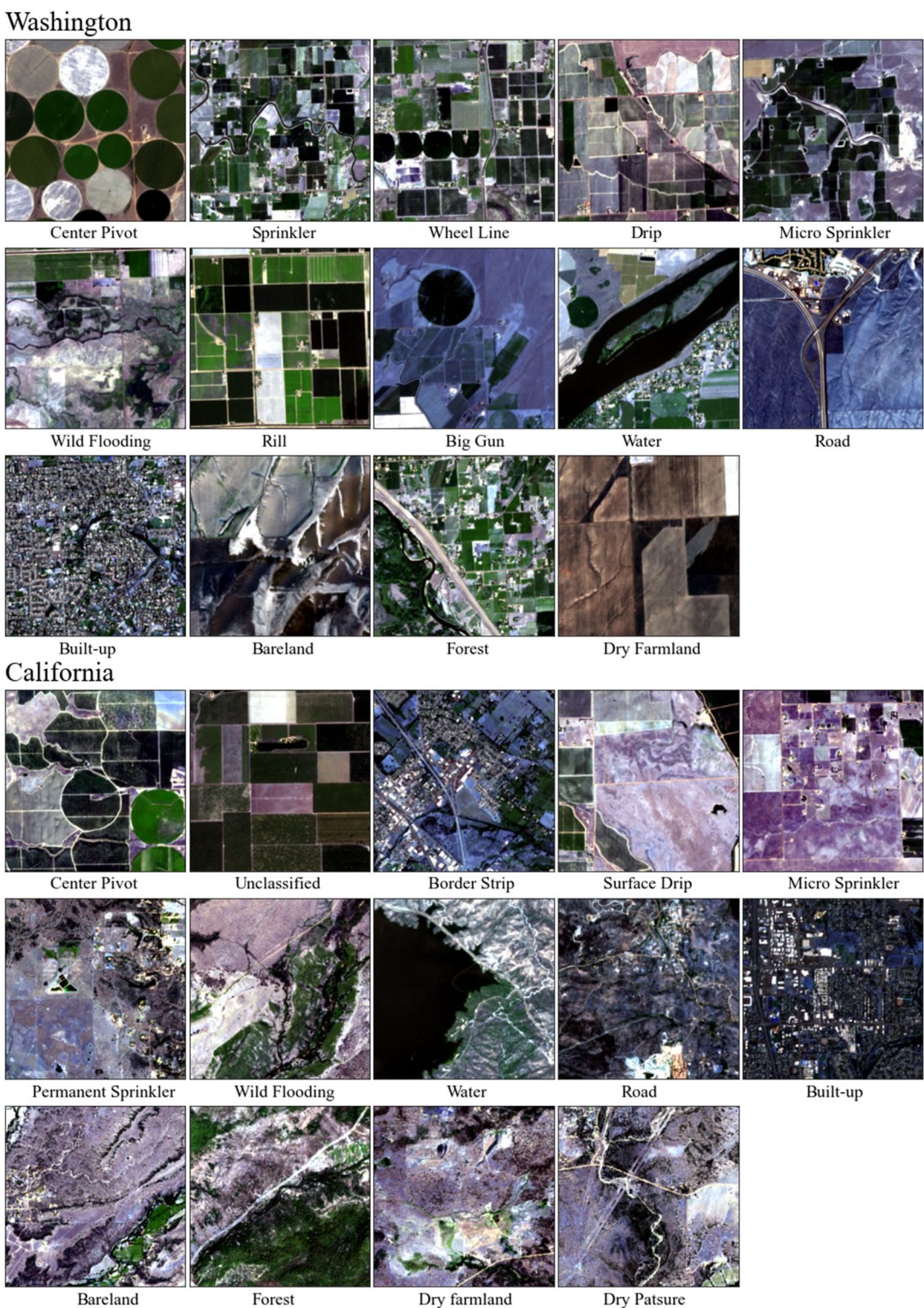

**Figure 2.** Representative exemplars of irrigation and non-irrigation.

In the state of Washington, the Washington state department of Agriculture (WSDA) developed a statewide agricultural land geodatabase (open accessed on the website: https://agr.wa.gov/departments/land-and-water/natural-resources/agricultural-land-use, accessed on 5 April 2023) [56]. WSDA mapping experts compiled crop data by applying fieldwork combined with the knowledge of crop identification skills and agricultural practices. These experts used vehicles and GPS-equipped laptops for fieldwork. WSDA measured the lands for cultivation and tracked the lands no longer used for agricultural

production. WSDA also utilized another source of land use data, coming from NASS Cropland Data Layer (CDL), to identify agricultural land use. The irrigation categories of WSDA crop data include Center Pivot, Sprinkler, Wheel Line, Drip, Micro Sprinkler, Wild Flooding, Rill, and Big Gun (Figure 2). WSDA updates the agricultural areas every 2–4 years. The geodatabase only provides irrigation information for the most recent year in which the data has been updated. Until 2023, we were only able to obtain the irrigation dataset for 2020.

In the state of California, land IQ, an institute that can provide services including responding to challenging agricultural problems and environmental problems, was contracted by the Department of Water Resources (DWR) of California to develop a statewide land use geodatabase for the Water Year. The land use geodatabase covers both the urban extent and the irrigable agriculture on a field scale (open accessed on the website: https://data.cnra.ca.gov/dataset/statewide-crop-mapping, accessed on 3 April 2023) [57]. Land IQ integrates knowledge of crop produce with ground truth information and multiple satellite and aerial image resources to conduct land use analysis on the field scale. The ground-truth data are obtained from the counties of Siskiyou, Modoc, Lassen and Shasta. Because the boundaries of homogeneous crop types, rather than legal parcel boundaries, represent the actual irrigable area, the experts of Land IQ used a crop category legend and a more specific crop type legend to classify individual fields. DWR staff determined the detailed review and revision of individual fields. Until 2023, the geodatabase provided irrigation information for the Water Years of 2018, 2019, 2020, and 2021. We adopted the data from 2019 to train the models and the data from 2018, 2020, and 2021 to test the extraction effect of the deep models in other years.

We used a subset of the validation dataset created by Ketchum et al. [60] as additional validation data. The dataset consisted of vector points delimiting sample regions for four classes: irrigated, dryland agriculture, uncultivated land, and wetlands. All field labels were created through manual interpretation of satellite images, National Agricultural Imagery Program (NAIP) imagery, and field surveys.

### 2.4. Daymet V4 (Daily Surface Weather Data on a 1-km Grid for North America, Version 4) Precipitation Data

The Daymet V4 dataset [61] provides gridded estimates at a 1 km × 1 km spatial resolution and a daily temporal resolution of the daily weather parameters for North America, Hawaii, and Puerto Rico. Daymet variables include minimum temperature, maximum temperature, precipitation, shortwave radiation, vapor pressure, snow water equivalent, and day length. In this study, we computed and downloaded cumulative spatial rainfall maps daily between 1 February and 31 October of each year with GEE. The rainfall data were not used in the mapping process. They were used to analyze and discuss the extraction results related to rainfall registration.

## 3. Methodology

### 3.1. Overview

Land cover types and surface landscapes are often complex in large-scale areas. First, traditional machine learning methods normally manually select features that match the target objects (known as feature engineering), and then apply classifiers to obtain extraction results. By comparison, DCNNs could adaptively learn representative and discriminating semantic features from training data hierarchically; hence, they have powerful discriminative power. This study utilized deep convolutional neural networks (DCNNs) with robust generalization capabilities to extract data on irrigated farmland from satellite imagery captured by the sentinel-2 satellite. Determination of irrigation at the field scale can be achieved by employing CNN models to process and analyze image information. Figure 3 displays a technical diagram, which includes the following parts: (1) collecting irrigation training samples; (2) inputting the training samples into the models to train

them for irrigation classification; and (3) using the trained models to classify the target images to generate an irrigation layer map.

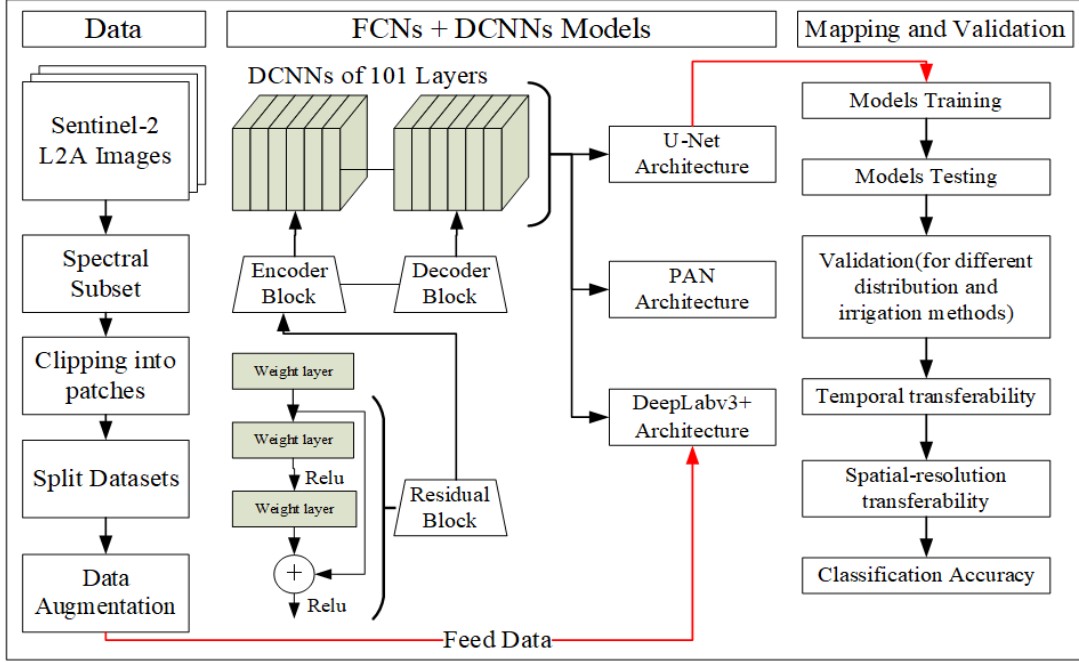

**Figure 3.** The flowchart of irrigation mapping using three FCNs + DCCNs models and Sentinel-2 L2A images.

### 3.2. Model Design

Several well-performing models, such as U-net [50], PAN [62], and DeepLabv3+ [63], have successfully applied the architecture of fully convolutional networks (FCNs) to various tasks, including traditional visual classification and land cover classification of remote sensing images. These models are called deep models because they utilize DCNNs as the underlying framework for feature extraction of the target objects, and they employ transpose convolution or interpolation operations as up-sampling layers to restore the map sizes of the inputs. In this study, three models were constructed by implementing the architectural framework of the models mentioned above. Due to the extensive coverage of the study areas and the availability of ample training data, we utilized the ResNet-101, a deep neural network architecture with 101 layers, as the backbone, also known as the encoder block in the FCN architecture. To simplify the models and computational requirements, we utilized bilinear interpolation operation instead of transpose convolution for the up-sampling layers, commonly known as the decoder block in the FCNs' architecture. In the context of the backbone, our approach involves prioritizing the preservation of location information and minimizing sparsity in the output features. To achieve this, we discarded pooling layers and applied dilation convolution layers to expand the perceptual field. The utilization of FCNs involves the substitution of fully connected layers with convolutional layers. Consequently, it is possible to input image patches of varying sizes into the model. The predicted image patches produced by FCNs are the same size as the input patches due to the up-sampling operation. Furthermore, the model assigns a class to each pixel of the output patches.

In an image, an object is typically represented by a cluster of pixels which can exhibit abstract features that individual pixels cannot convey. The behavior of the encoder block that extracts features of the object can be described as follows. A convolutional layer processes the image by sliding a window of the weight matrix over it. The weight matrix multiplies with the pixels in the window's area, and the sum of the product represents the feature which the convolutional layer has learned. All the feature points form the feature

map, which is then fed into the next convolutional layer. Consequently, the convolutional layer could model the spatial relationship between neighboring pixels. In the region covered by the window, all the pixels represent each pixel's local neighborhood information, also known as contextual information. The region of the input image connected with the feature point is known as the receptive field. In the feature map of a certain layer, the receptive field of the feature point covers a large area of the input image. Therefore, the convolutional layer could model all the pixels in the image patch. CNN could learn and characterize the similarities and differences of all the pixels in an image to obtain the abstract features of objects and backgrounds. We set the input patches to have image sizes of $256 \times 256 \times 36$ pixels and the output patches sizes of $256 \times 256 \times 1$.

### 3.3. Experiment Design

The difference between non-irrigated and irrigated fields may not be solely in terms of spectral information but also in the shape of the fields and surrounding context. CNN can use dense pixel-wise labels, while RF models (we use the most frequently used RF model as an example) pick and choose points for training. CNN could incorporate learned representations of images on multi-scales when predicting, picking up patterns and contextual information. RF models, in contrast, rely on handcrafted features that capture the spatial context or otherwise discard spatial information in the classification.

We generated the samples using the irrigated farmland datasets as labels and the corresponding Sentinel-2 L2A images as input images for the deep models. The training samples should contain as many land cover types as possible, including different irrigated farmland types (positive samples) and different non-irrigated farmland types (negative samples), such as center pivot irrigation, sprinkler irrigation, road, and built-up. As described in Section 2.2, we fed nine temporal median images (a total of 36 bands), sorted by time, into the models. When training deep models, the validation set is indispensable for preventing model overfitting, and the test set is essential for validating and assessing the performance of trained models. In our study, we thoroughly shuffled all available samples, subsequently allocating them to the training, validation, and test sets at a ratio of 7:2:1, respectively. Specifically, in Washington, we assigned 2743 samples to the training set, 783 to the validation set, and 382 to the test set. Similarly, in California, our allocations resulted in 4060 samples in the training set, 1160 in the validation set, and 580 in the test set. Based on the test set and the validation dataset created by Ketchum et al. [60], we calculated the evaluation metrics of the deep models.

To increase the amounts of training data in order to improve the applicable ability of the models, we usually need to perform data augmentation of samples in both the training and validation sets. We applied four augmentation methods, including the horizontal flip, the vertical flip, the random rotate of 90°, and the transpose. When feeding the images into the models, we randomly adopted each method for the images with a probability of 0.5. We set the batch size to 32.

In order to accelerate the training process, we employed the transfer learning strategy. The weights of all three networks were initialized using the parameters of pre-trained models trained on the ImageNet dataset. As a benchmark in image classification, the ImageNet dataset contains over 14 million annotated images that cover one thousand classes. The large learning rate and the large training epochs were not needed, and we set the initial learning rate as 0.0005 and the total number of epochs as 120. Along with the iteration of training epochs, we usually fine-tune the learning rate to prevent gradient explosion and to accelerate the training process. At one training epoch, we applied the Adam algorithm to optimize the gradient descent and weight decay to 0.00001 to avoid overfitting, and the momentum to 0.9 to regularize the learning. Meanwhile, we reduced the learning rate iteratively by 0.2 after every ten epochs. At one training epoch, we calculated the accuracy metrics on the validation set using the models trained on the training set. Moreover, if the value of the specific metric on the validation set of this epoch was greater than the previous epoch, the model weight of this epoch would be selected.

The model with the highest metric value on the validation set among 120 training epochs was considered the best. Then, we fed the images in the test set to the best model to calculate the output map. The output map was used to compare with the original label map corresponding to the input image and to calculate the evaluation metrics. Implementing both temporal and spatial resolution transfer of the deep models is similar to the previous evaluation process of the best model on the test set.

Quantifying the extent of irrigated croplands across different years is important for water use management [28,29]. However, this task could be difficult due to the variability of crop growth and the high cost associated with conducting field surveys. CNN could learn the abstract features of objects represented by a cluster of pixels and be less affected by individual pixels. Therefore, we sought to extract irrigation information from images captured in years other than 2019 using models trained on 2019 images, and evaluated the temporal portability of the deep models (in California only, because of the availability of multi-year irrigation layer datasets). We selected the images in each year, besides 2019, separately, covering the extent of the S6 location in Figure 1 (California: from 2018 to 2021) as the test set for every year. Concerning remote sensing images, high spatial resolution allows for detailed observation of the target objects, and high temporal resolution enables monitoring of changes to these objects over time. However, it is often difficult to find image products with high spatial and temporal resolution, and was especially so before the introduction of Sentinel images. For instance, Landsat images have a spatial resolution of 30 m and a temporal resolution of 16–18 days, while MODIS images have a spatial resolution of 250–1000 m and a temporal resolution of 1 day. To cater to different needs, we usually label images with varying spatial resolutions separately to obtain samples and train models, which is costly. As our eyes can roughly distinguish objects of different spatial resolutions, computer vision models can also do so. Leveraging the ability of CNN to characterize objects, we attempted to extract irrigation information from images with spatial resolutions other than 10 m by employing models trained on images of 10 m resolution. Furthermore, we examined the spatial resolution portability of these deep learning models. We constructed a test image set for each spatial resolution (7 test sets in total) by choosing images of other resolutions (20, 60, 90, 100, 200, and 300 m) in addition to 10 m resolution ones. Moreover, we obtained the truth labels corresponding to the test images of different spatial resolutions by resampling the original irrigated cropland data.

### 3.4. Accuracy Assessment

To evaluate the performance of the deep models and the effect of irrigation extraction, we selected the following accuracy indices, including Precision, Recall, F1-score, Intersection over Union (IoU), overall accuracy (OA), and Kappa, in this study. The formulae used to calculate these accuracy indices are summarized below:

$$\text{Precision} = \frac{\text{TP}}{\text{TP} + \text{FP}} \tag{1}$$

$$\text{Recall} = \frac{\text{TP}}{\text{TP} + \text{FN}} \tag{2}$$

$$\text{F1} - \text{score} = \frac{2\text{TP}}{2\text{TP} + \text{FN} + \text{FP}} \tag{3}$$

$$\text{OA} = \frac{\text{TP} + \text{TN}}{\text{TP} + \text{TN} + \text{FN} + \text{FP}} \tag{4}$$

$$\text{EA} = \frac{(\text{TP} + \text{FP}) \times (\text{TP} + \text{FN}) + (\text{FN} + \text{TN}) \times (\text{FP} + \text{TN})}{(\text{TP} + \text{TN} + \text{FN} + \text{FP})^2} \tag{5}$$

$$\text{Kappa} = \frac{\text{OA} - \text{EA}}{1 - \text{EA}} \tag{6}$$

where TP is the number of the irrigation pixels that are truly classified as irrigation, TN is the number of the irrigation pixels that are truly classified as non-irrigation, FP is the

number of non-irrigation pixels that are falsely classified as irrigation, FN is the number of irrigation pixels that are falsely classified as non-irrigation, and EA is the expected accuracy.

$$IoU = \frac{A \cap B}{A \cup B} \tag{7}$$

where both *A* and *B* are the collections of pixels. The IoU is used to describe the overlap extent of two boxes in which the values of pixels can represent the land classes. The IoU becomes greater as the overlap region grows.

## 4. Results

Vegetation indices (VIs), such as the normalized difference vegetation index (NDVI), enhanced vegetation index (EVI), and greenness index (GI), are widely used to identify irrigated croplands [17,18,63]. A maximum VI derived from the annual time series could be seen as a proxy for the peak level of photosynthetic activity, the highest biomass, and, possibly, the densest vegetation canopy [64]. The maximum annual peak VI for any crops could be attributed to the consistent adequate soil moisture delivered by irrigation during the growing season [63]. Usually, irrigated crops have higher peak VIs than non-irrigated crops [63,65]. In this study, the annual median composites of sentinel-2 bands 2–4 (blue, green, red) and the annual maximum composites of EVI were computed from the sentinel-2 data. We visualized the median composites of sentinel-2 bands 2–4 and maximum composites of EVI to help to analyze the extraction results of the deep models. We computed the spatial mean values of daily cumulative rainfall maps of the following examples as the precipitation values of the corresponding regions.

### 4.1. Model Accuracy

We used the trained model to predict the results for regions of the selected samples (Figure 4).

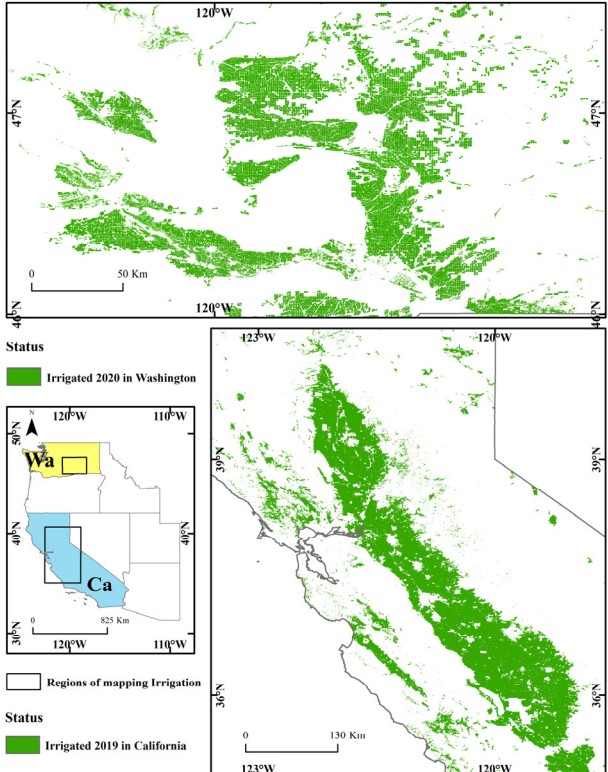

**Figure 4.** Irrigation status for the year 2020, as predicted by DeepLabV3+ in Washington, and for the year 2019, as predicted by U-net in California, at 10 m resolution.

Tables 1 and 2 summarize the quantitative evaluation metrics of the three deep models for irrigation extraction. Table 1 shows that good classification results were achieved for all three deep models in both state domains, with F1-scores around 0.94 (the lowest values were 0.88 for IoU and 0.91 for Kappa). In Washington, the highest extraction accuracy was achieved by DeepLabV3+, and the lowest by PAN. In California, the highest extraction accuracy was achieved by U-net, and the lowest by DeepLabV3+. Table 2 shows that the classification accuracy was higher based on the validation dataset (vector points) created by Ketchum et al. [60].

**Table 1.** Quantitative indices of the classification (irrigation extraction) of different networks: precision, recall, F1-score, IoU, OA, and Kappa for Washington and California.

| | Washington | | | | | | California | | | | | |
| | Precision | Recall | F1-score | IoU | OA | Kappa | Precision | Recall | F1-Score | IoU | OA | Kappa |
|---|---|---|---|---|---|---|---|---|---|---|---|---|
| DeepLabV3+ | 0.94 | 0.95 | 0.95 | 0.90 | 0.98 | 0.94 | 0.93 | 0.94 | 0.94 | 0.88 | 0.96 | 0.91 |
| PAN | 0.94 | 0.94 | 0.94 | 0.89 | 0.98 | 0.93 | 0.94 | 0.95 | 0.94 | 0.89 | 0.97 | 0.92 |
| U-net | 0.94 | 0.95 | 0.94 | 0.89 | 0.98 | 0.93 | 0.95 | 0.95 | 0.95 | 0.90 | 0.97 | 0.92 |

**Table 2.** Quantitative indices of the classification (irrigation extraction) based on the validation dataset created by Ketchum et al.: precision, recall, F1-score, IoU, OA, and Kappa for Washington and California.

| | Washington | | | | | | California | | | | | |
| | Precision | Recall | F1-score | IoU | OA | Kappa | Precision | Recall | F1-Score | IoU | OA | Kappa |
|---|---|---|---|---|---|---|---|---|---|---|---|---|
| DeepLabV3+ | 0.98 | 0.98 | 0.98 | 0.96 | 0.97 | 0.94 | 0.98 | 0.93 | 0.96 | 0.92 | 0.93 | 0.91 |
| PAN | 0.98 | 0.98 | 0.98 | 0.96 | 0.98 | 0.94 | 0.98 | 0.92 | 0.95 | 0.91 | 0.92 | 0.92 |
| U-net | 0.98 | 0.98 | 0.98 | 0.95 | 0.97 | 0.93 | 0.98 | 0.93 | 0.96 | 0.92 | 0.93 | 0.92 |

*4.2. Irrigation Extraction results with Different Distribution States*

Figure 5 shows the extraction effect of irrigation based on the deep models in the study areas. Table 3 displays the cumulative daily precipitation data from 1 February to 31 October for four selected parcels of land, namely, S1–S4. These parcels were chosen to represent different characteristics, such as the presence or absence of irrigation and the differentiation between rain-fed and irrigated land. Additionally, the parcels were selected to showcase variations in field distribution, with S1 and S3 representing concentrated and regularly distributed fields and S2 and S4 representing discrete and highly heterogeneous fields. It can be seen that these four areas displayed different spectral, textural, and morphological characteristics, and that the three deep models performed well in both of the study regions. Irrigated fields could be effectively identified whether or not crops were growing in the fields. The bare lands scattered throughout the four landscapes were also well detected. In terms of different shapes, irrigated fields with concentrated and regular distribution shape could be distinguished from other land classes, such as rivers, lakes, roads, buildings, bare land, forests, rainfed cropland, and rainfed grazing land (S1 and S3 in Figure 5), even though they possessed similar spectral features in different types of land cover (S1 and S3 in California). Irrigated and rainfed croplands with similar spectral, textural, and morphological features are mostly distinguishable as well (S3 and S4 in Figure 5). When it comes to different backgrounds, irrigation is much easier to detect from non-croplands than from rainfed croplands, which is in line with a wide range of understanding (S1 and S3 in Figure 5). The reason is clear: similar spectral and textural features interfere with the differentiation between irrigation and other croplands. Discrete and heterogeneous distribution patterns make it difficult to distinguish irrigation from non-cropland backgrounds or rainfed-cropland backgrounds (S2 and S4 in Figure 5). In regions where multiple land classes are interspersed, forests, villages, and rainfed farmland could be filtered out (S2 and S4 in California); however, some misclassifications still

occurred. Small plots and buildings with jagged and staggered features were frequently misclassified (S2 in Washington and S4 in California), even though there are obvious spectral differences between irrigation and other land types (S2 in Washington). In areas with greater cumulative precipitation, usually, land cover is greener and there are smaller spectral differences between irrigation and other land types (Table 3 and Figure 5), which also interferes with the distinction between irrigation and other land types (S1–S4 in California).

**Table 3.** Daily cumulative precipitation for the period between 1 February and 31 October for S1–S4 in Washington and California (mm).

| | Washington | | | | California | | | |
|---|---|---|---|---|---|---|---|---|
| **Examples** | **S1** | **S2** | **S3** | **S4** | **S1** | **S2** | **S3** | **S4** |
| Precipitation | 164.6 | 100.25 | 135.9 | 109.56 | 584.26 | 973.57 | 155.49 | 421.74 |

(**A**) Washington

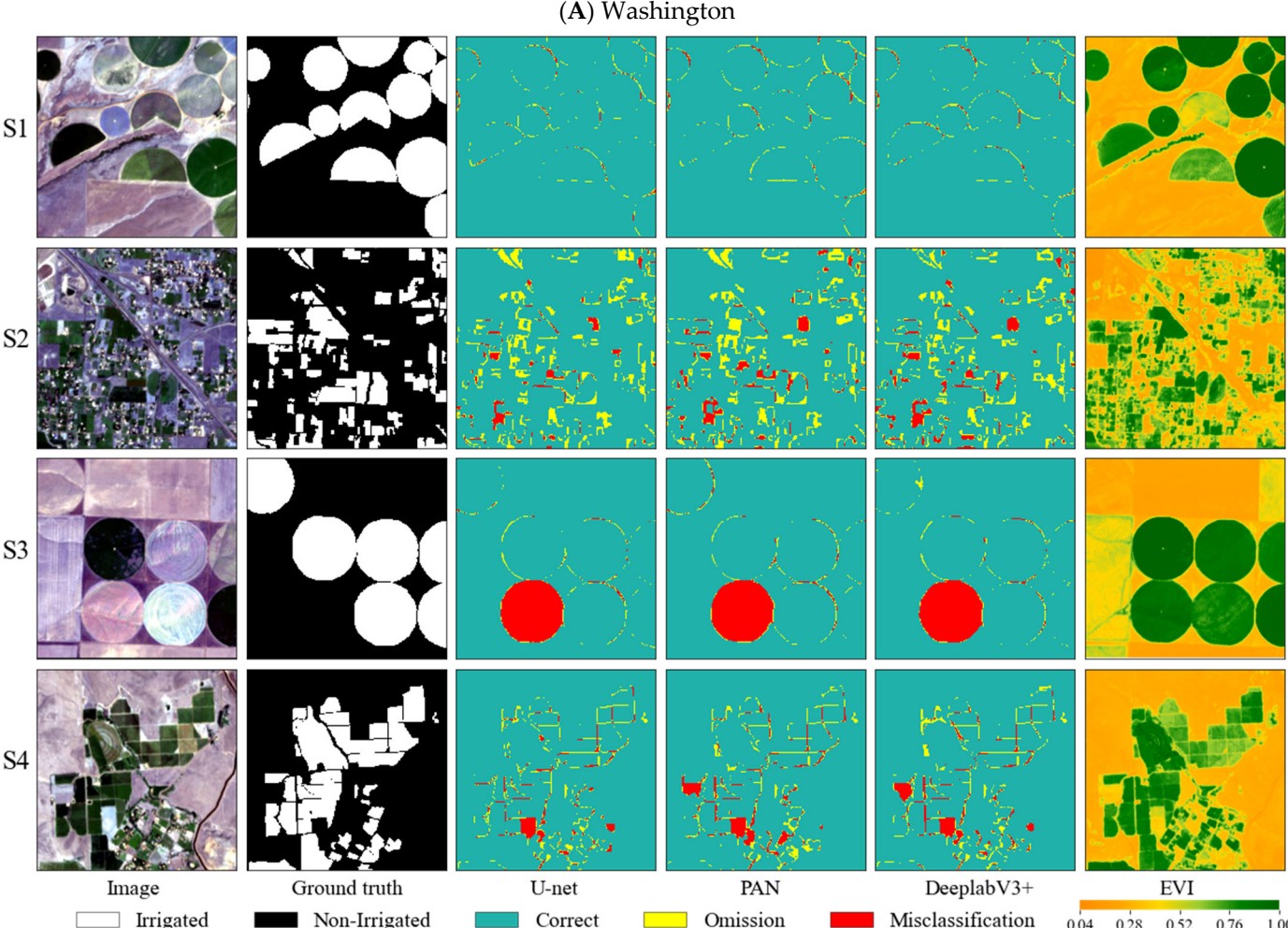

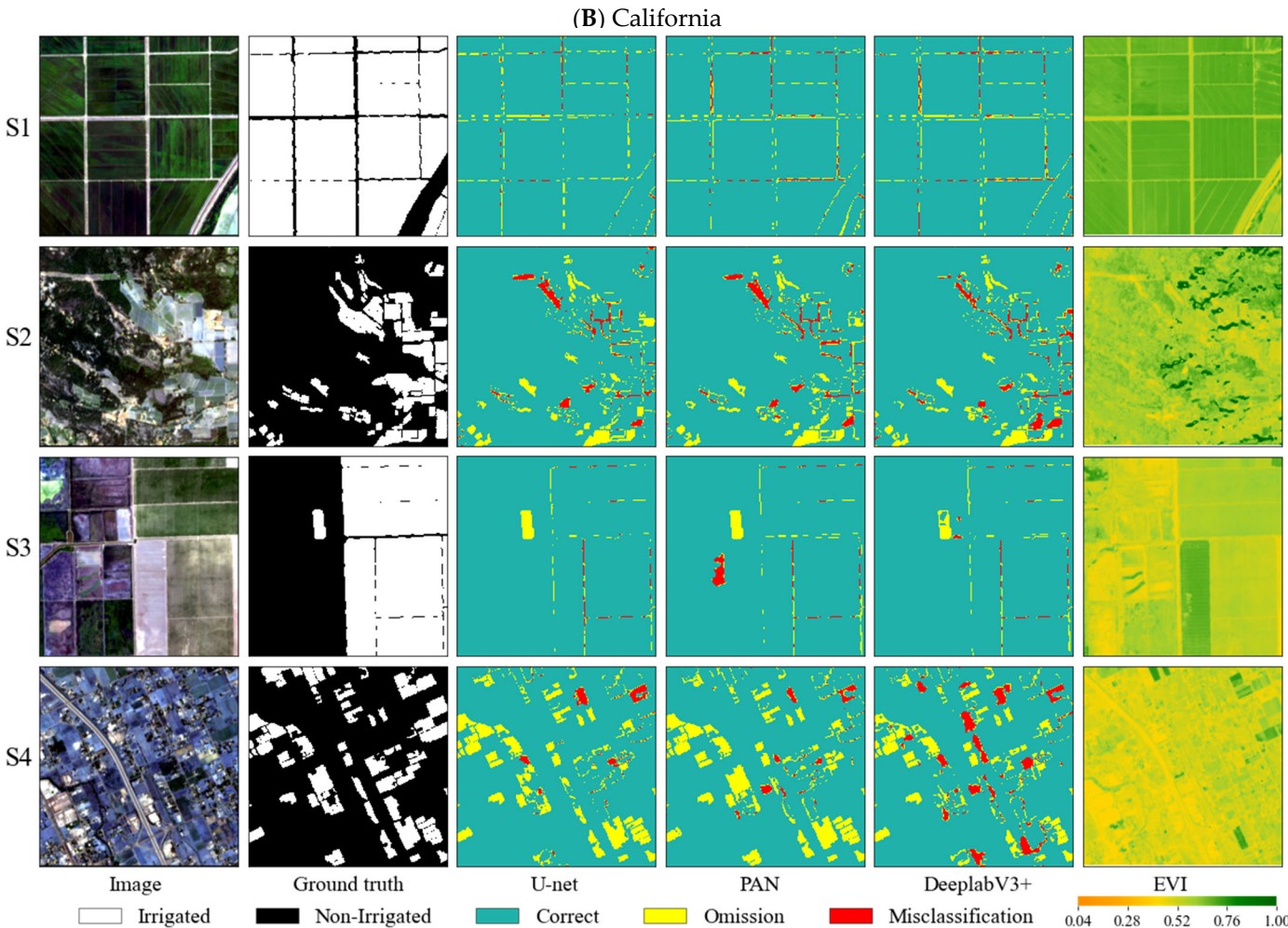

**Figure 5.** Irrigation extraction effect based on different models for (**A**) Washington and (**B**) California. Distinguishing between irrigation and non-cropland (S1 and S2); irrigation and non-irrigated land (S3 and S4); concentratedly and regularly distributed landscapes (S1 and S3); and fragmented and heterogeneous landscapes (S2 and S4). EVI was added to evaluate the extraction effect. Typical subsets of 2.56 × 2.56 km² in the study areas depict samples of the mapping results. The examples correspond to the marks in the Figure 1 location map.

### 4.3. Irrigation Extraction Results among Fields Served by Different Types of Irrigation

Different irrigation types result in varying appearances and features of fields, such as the circular shape of fields irrigated by center pivot systems. To investigate whether irrigation extraction is affected by irrigation methods, we examined the extraction results of the deep models across fields served by different types of irrigation (Figure 6). Table 4 presents the daily cumulative precipitation for (1)–(8) in Washington and (1)–(5) in California.

(**A**) Washington

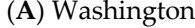

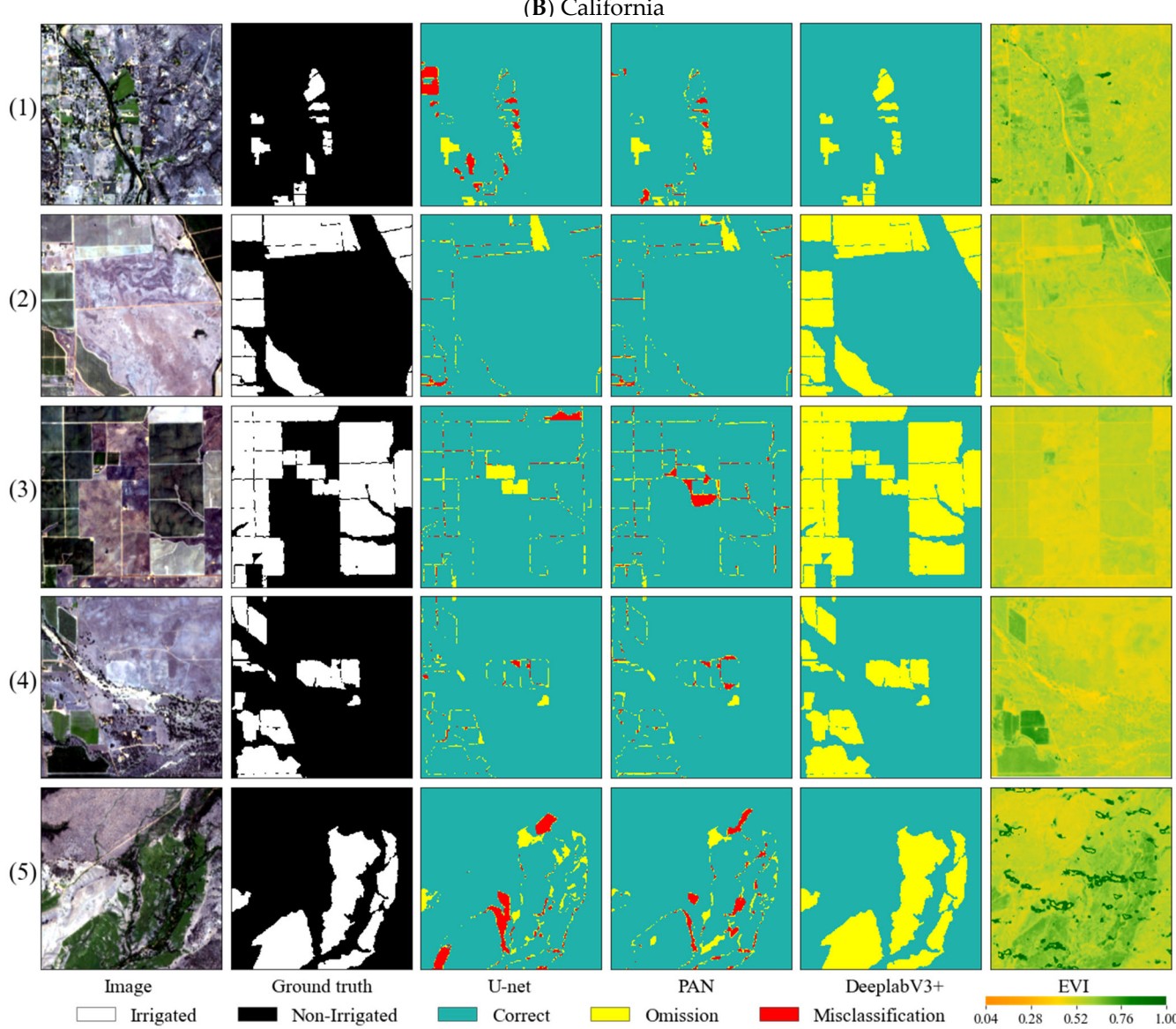

**Figure 6.** Irrigation extraction effect among the fields served by specific irrigation methods based on different models for the study areas. EVI is added to evaluate the extraction effect. There are 8 irrigation categories in Washington (**A**), including Center Pivot (1), Sprinkler (2), Wheel Line (3), Drip (4), Micro Sprinkler (5), Wild Flooding (6), Rill (7), and Big Gun (8). There are 5 irrigation categories In California (**B**), including Border Strip (1), Surface Drip (2), Micro Sprinkler (3), Permanent Sprinkler (4), and Wild Flooding (5). Typical subsets of 2.56 × 2.56 km² depict samples of the mapping results. The examples correspond to the marks in the Figure 1 location map.

In Washington, we achieved ideal extraction outcomes in the fields served by the Center Pivot (1), Sprinkler (2), Drip (4), Micro Sprinkler (5), and Rill (7) systems, with the exception of gaps in certain plots. These results were primarily achieved due to the well-distributed placement of the plots and their distinct characteristics compared to other land cover types. There were numerous misclassifications and omissions in fields served by Wheel Line (3) systems, which typically irrigate small fields interspersed among other land classes. In the fields served by Wild Flooding (6) systems, DeepLabV3+ demonstrated the best extraction effects, while PAN yielded the most omissions. This irrigation type is distinct from others and is commonly used for pastures and pasture crops, allowing water to flow freely onto plots where the land is not tidied. The fields served by Big Gun (8) systems experienced many omissions in the extraction results of U-net and numerous

misclassifications of DeepLabV3+. This irrigation method is employed in plots of almost any shape, as well as areas where the use of other methods proves challenging, particularly in areas where it is difficult to relocate irrigation facilities once the crops have matured, such as sugar cane and corn. In areas with less cumulative precipitation, there are often greater spectral differences between irrigation and other land types (Table 4 and Washington in Figure 6). Different spectral features help to identify irrigation, while the distribution patterns that small fields intersperse among multiple land classes interfere with the distinction between irrigation and other land classes ((1)–(8) in Washington).

**Table 4.** Daily cumulative precipitation for the period between 1 February and 31 October for (1)–(8) in Washington and (1)–(5) in California (mm).

| | Washington | | | | | | | | California | | | | |
|---|---|---|---|---|---|---|---|---|---|---|---|---|---|
| Examples | (1) | (2) | (3) | (4) | (5) | (6) | (7) | (8) | (1) | (2) | (3) | (4) | (5) |
| Precipitation | 127.11 | 187.48 | 265.90 | 95.60 | 91.85 | 80.63 | 92.65 | 149.20 | 699.08 | 597.81 | 597.69 | 600.20 | 830.91 |

In California, DeepLabV3+ exhibits a significantly inferior performance in comparison to U-net and PAN in terms of the extraction effect for fields served by the five types of irrigation, with DeepLabV3+ displaying many obvious omissions. The ideal extraction effect for U-net and PAN was observed in the fields served by the Border Strip (1), Surface Drip (2), Micro Sprinkler (3), and Permanent Sprinkler (4) systems, which can be attributed to the characteristics of these fields. For Border Strip (1), the fields are divided into strips through the construction of dikes or monopolies, which cause water to flow in sheets to the strips. Surface Drip (2) is extensively employed for the irrigation of perennial crops (trees and vines) and annual row crops. The Permanent Sprinkler (4) system is suitable for almost all dryland crops and is widely used for field crops, cash crops, vegetables, and garden meadows in plains and mountainous areas. It does not produce surface runoff, and the moisture on the ground is uniform after irrigation. As for the fields served by Wild Flooding (5), U-net and PAN can accurately extract the interior regions of the fields, although misclassifications and omissions exist at the outer edge and in locations where the irrigated areas meet other land classes. In areas with greater cumulative precipitation, there are often similar spectral features between irrigation and other land types (Table 4 and Figure 6), which interferes with the distinction between irrigation and other land types ((1)–(5) in California). Different textural and morphological features help to identify irrigation.

*4.4. Assessment of Temporal Portability of the Deep Models to Extract Irrigation*

A fundamental issue of mapping irrigation at large scale is the portability of the models. Temporal portability refers to the ability of a model to generalize data from another time that has not been exposed to training data, which means that trained models from a specific time are applied to other time periods.

Figure 7 displays the outcomes of extracting irrigation for different years using models trained on the 2019 data of California. All examples were located in the same sample site. The amounts of precipitation for S5 in 2018–2021 were 62.05, 152.05, 115.48, and 52.14 mm, respectively. Small differences in precipitation affected the greenness of land covers less across the years (Figure 7). It is apparent that spectral and textural features vary based on the planting and growing conditions of crops across the years. U-net exhibited a significantly inferior performance in comparison to PAN and DeepLabV3+, with U-net displaying many obvious omissions. The ideal extraction effects for PAN and DeepLabV3+ were observed in different years. Irrigated fields could be identified, although with similar spectral features in different land classes (2021). Misclassifications on small plots interspersed throughout different land classes suggest that similar distribution patterns affect the models' classification performances across years.

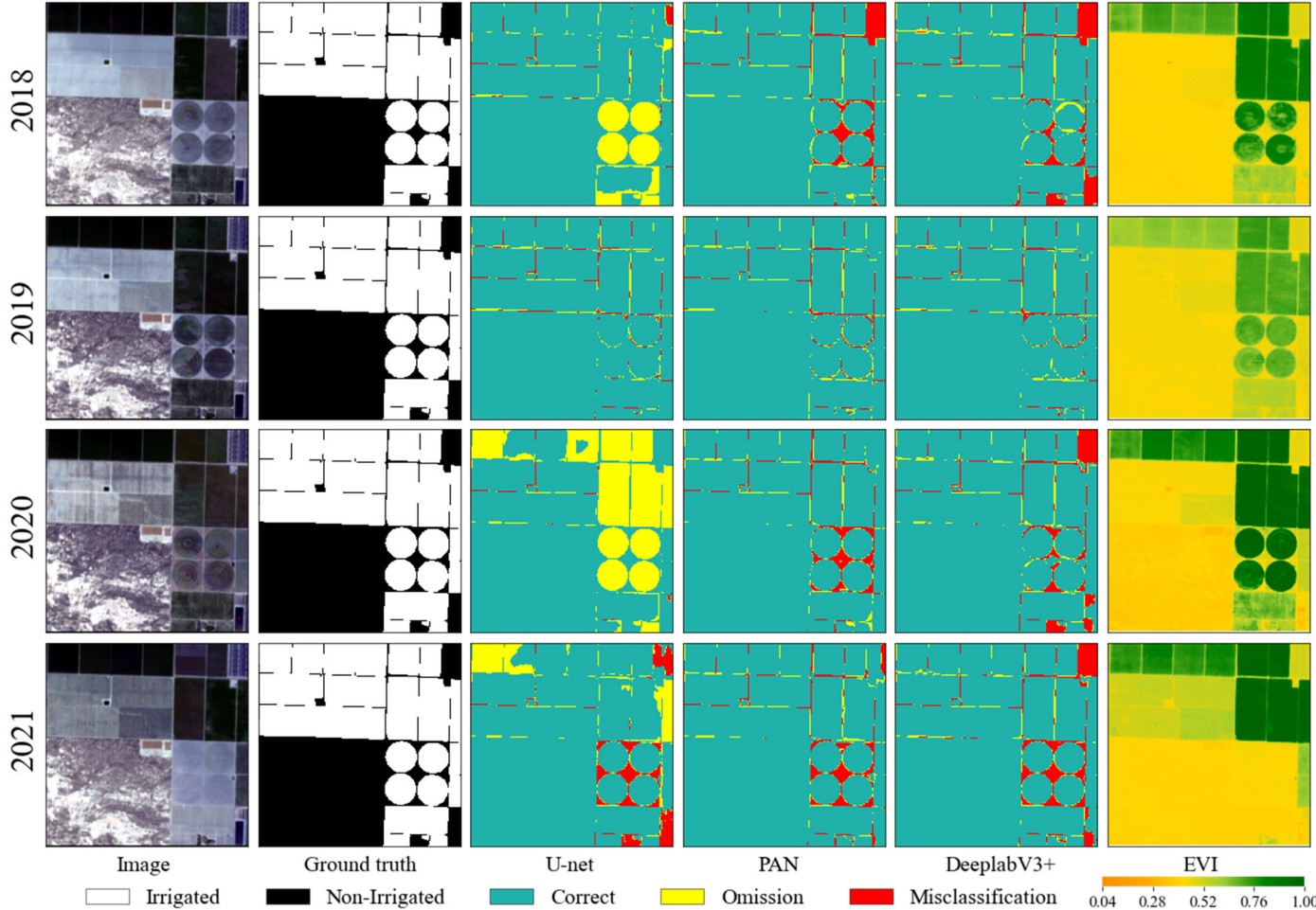

**Figure 7.** The examples of classification's effect of distinguishing between irrigation and non-irrigation in different years using the models trained by data acquired in 2019 for California. EVI was added to evaluate the extraction effect. Typical subsets of 2.56 × 2.56 km² depict samples of the mapping results. The examples correspond to S5 in the Figure 1 location map.

Figure 8 shows the quantitative evaluation metrics for the classification performance of the deep models in different years. The amounts of precipitation for T11SKV in 2018–2021 were 116.49, 230.51, 141.74, and 66.47 mm, respectively. The evaluation results from 2018 to 2021 indicate that for all models, the lowest value of Recall was 0.91, the lowest value of OA was 0.87, and the lowest values of other metrics were 0.73. This indicates that irrigation could be identified correctly in most cases in different years, and only a small number of irrigated fields were omitted. The interval between the prediction year and the training year was longer, while the accuracy in the prediction year was lower. This may be because the time interval was longer and the environmental change was greater. By comparing PAN and DeepLabV3+ in 2020 and 2018, we found that the accuracy was higher in predicted years with more precipitation. This may have been due to more precipitation in training years; the predicted years with more precipitation showed results similar to those of the training years. Among the three models, PAN exhibited the best generalization ability, while U-net displayed the worst, across different years.

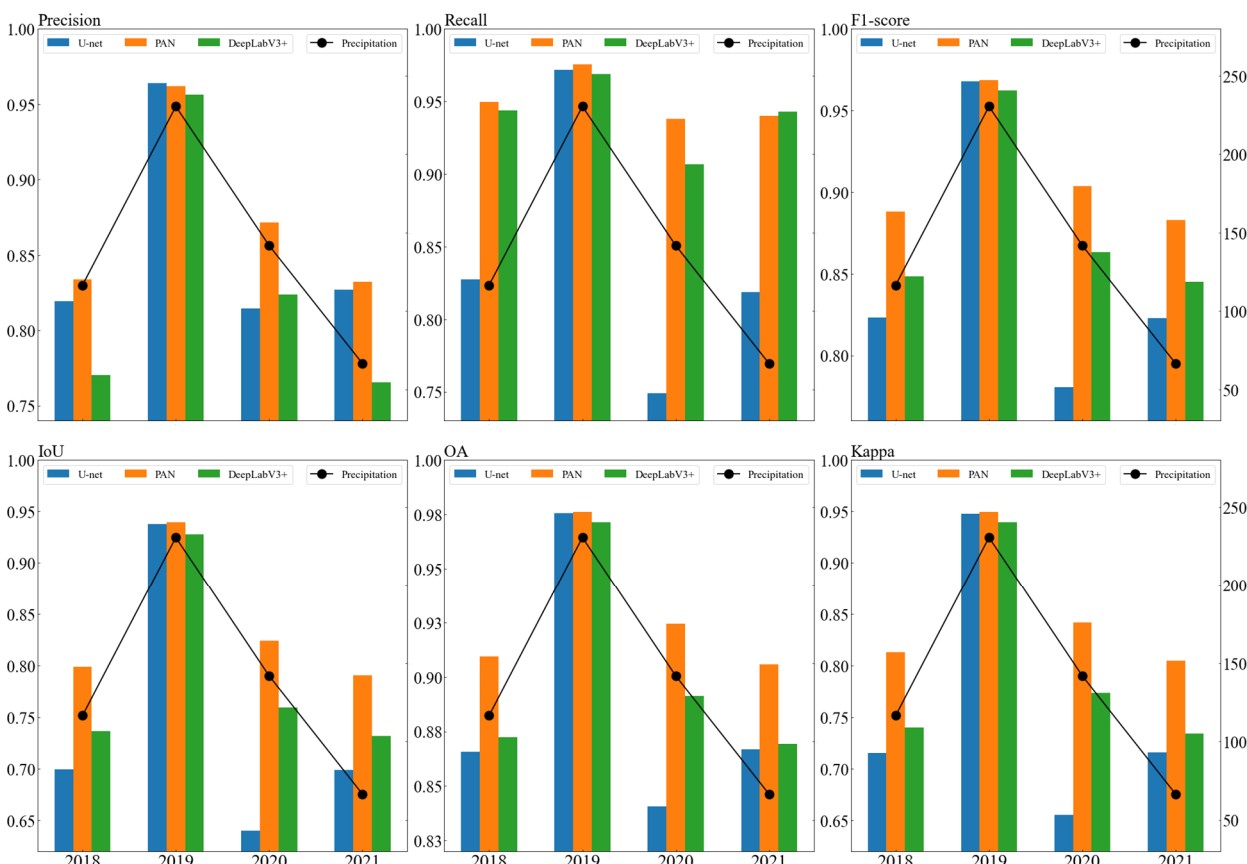

**Figure 8.** The quantitative evaluation of distinguishing between irrigation and non-irrigation in other years based on the models trained by data acquired in 2019 for California. The indices include Precision, Recall, F1-score, IoU, OA, and Kappa.

### 4.5. Assessment of Spatial Resolution Portability of the Deep Models to Extract Irrigation

Spatial resolution portability is the ability of a model to generalize data from the same time, but with different spatial resolution, at a given location, which means that trained models of a specific resolution are applied to other resolutions. Figure 9 shows the irrigation extraction results with varying spatial resolutions (20, 60, 90, 100, 200, and 300 m) based on deep models trained on 10 m resolution data in July. For visualization, we selected one tile of the image for each area (T10TGS for Washington and T11SKV for California). The amounts of precipitation in T10TGS and T11SKV were 98.17 and 230.51 mm, respectively. As the spatial resolution decreased, the details of objects in the images became coarse, and the spectral features and texture characteristics differed. The deep models' adaptations to different resolutions was demonstrated by an increase in the difference and decline in the extraction effect as the resolution difference between the validation and training data increased (R10 and R20 compared with R20 and R60 of Washington and California in Figure 9). In Washington, the number of misclassifications in the extraction results increased marginally with the decreasing spatial resolution, while the number of omissions increased significantly. This indicates that the deep models' ability to identify irrigation decreased considerably, but less so for non-irrigation. The extraction effect declined sharply as the spatial resolution changed from 20 to 60 m, but there was no noticeable change from 20 to 100 m. In California, a significant change occurred from 100 to 200 m. The number of omissions and misclassifications in the extraction results increased as the spatial resolution decreased from 10 to 100 m. At 200 m, almost all classifications were misclassified, and at 300 m, there were misclassifications for U-net and PAN and omissions for DeepLabV3+.

(**A**) Washington

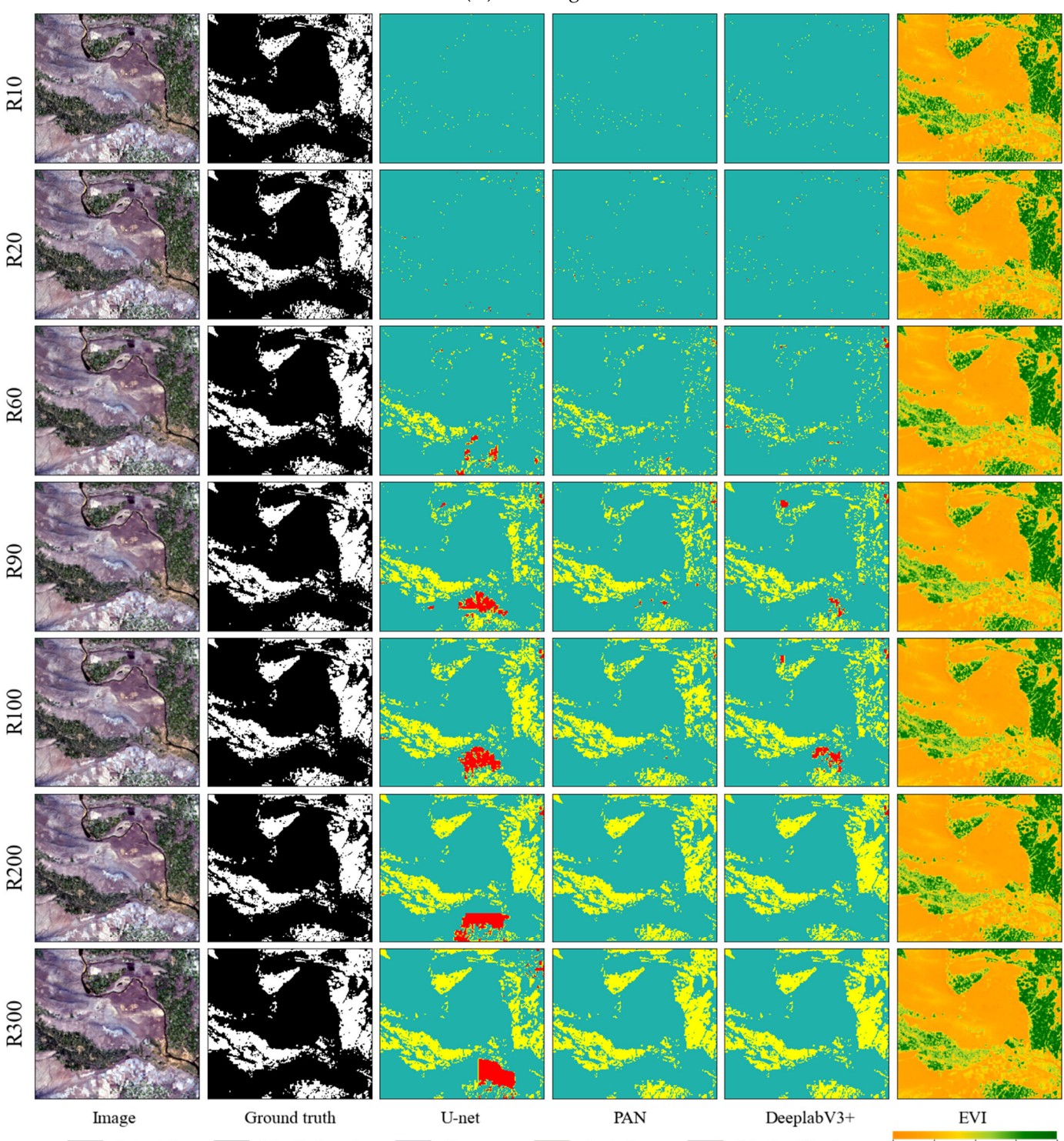

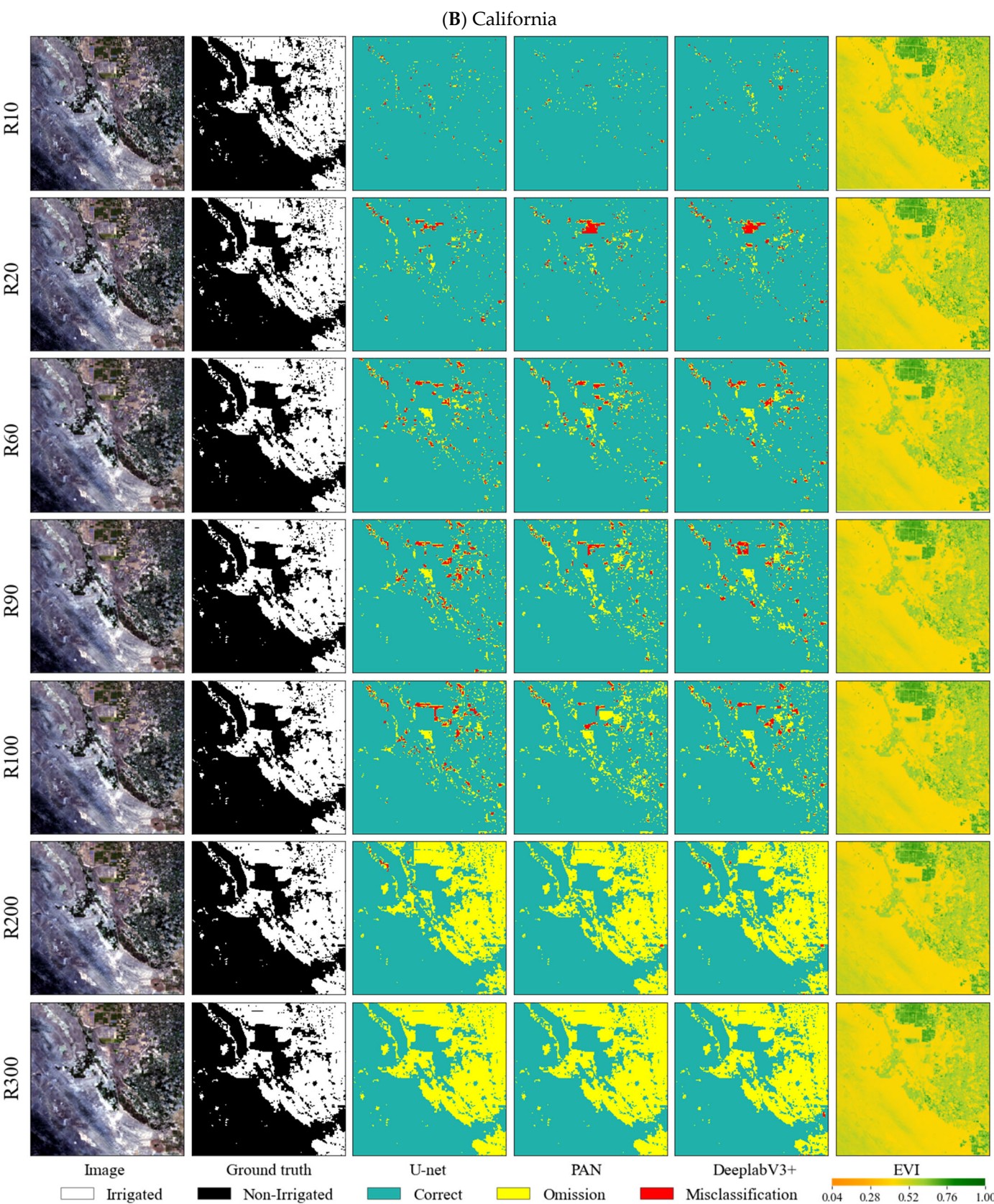

**Figure 9.** Examples of the classification effect of distinguishing between irrigation and non-irrigation at different spatial resolutions based on the trained models using data of a 10 m resolution. EVI was added to evaluate the extraction effect. The spatial resolution changed from 10 to 300 m for (**A**) Washington and (**B**) California. Typical subsets of 109.8 × 109.8 km² depict samples of the mapping results. The examples correspond to S6 in the Figure 1 location map.

Figure 10 displays the quantitative metrics measuring the classification performance of the deep models at various spatial resolutions. The values of these metrics decreased as the spatial resolution decreased in both study areas. The worst performance occurred at 300 m resolution, except for U-net, which showed the worst performance at 100 and 200 m resolution. The values of these metrics decreased as the spatial resolution decreased in both study areas. The worst performance occurred at 300 m resolution, except for U-net, which had the worst performance at 100 and 200 m resolution. In Washington, the values of all metrics varied very little from 10 to 20 m resolution. From 10 to 60 m resolution, the lowest value of OA was 0.87, that of IoU was 0.52 (excluding U-net), and that of the other metrics was 0.56. U-net demonstrated stability across different spatial resolutions, whereas PAN did not exhibit the same relative stability. The models' metrics declined rapidly from 20 to 90 m resolution, and their accuracies trended towards stability from 90 to 300 m. In California, the lowest value of Precision and OA was 0.85, that of F1-score was 0.70, that of Kappa was 0.60, and that of Recall and IoU was 0.54 from 10 to 100 m resolution. From 10 to 300 m resolution, the lowest value of Precision was 0.81 (excluding DeepLabV3+), and that of OA was 0.68. U-net achieved the best transferability/portability performance, whereas PAN exhibited the worst relative performance. Unlike in Washington, the models' metrics remained relatively stable from 20 to 100 m resolution, and their metrics declined rapidly from 100 to 300 m resolution. The generalization performance of the models at different spatial resolutions was better in the region with less precipitation.

(**A**) Washington

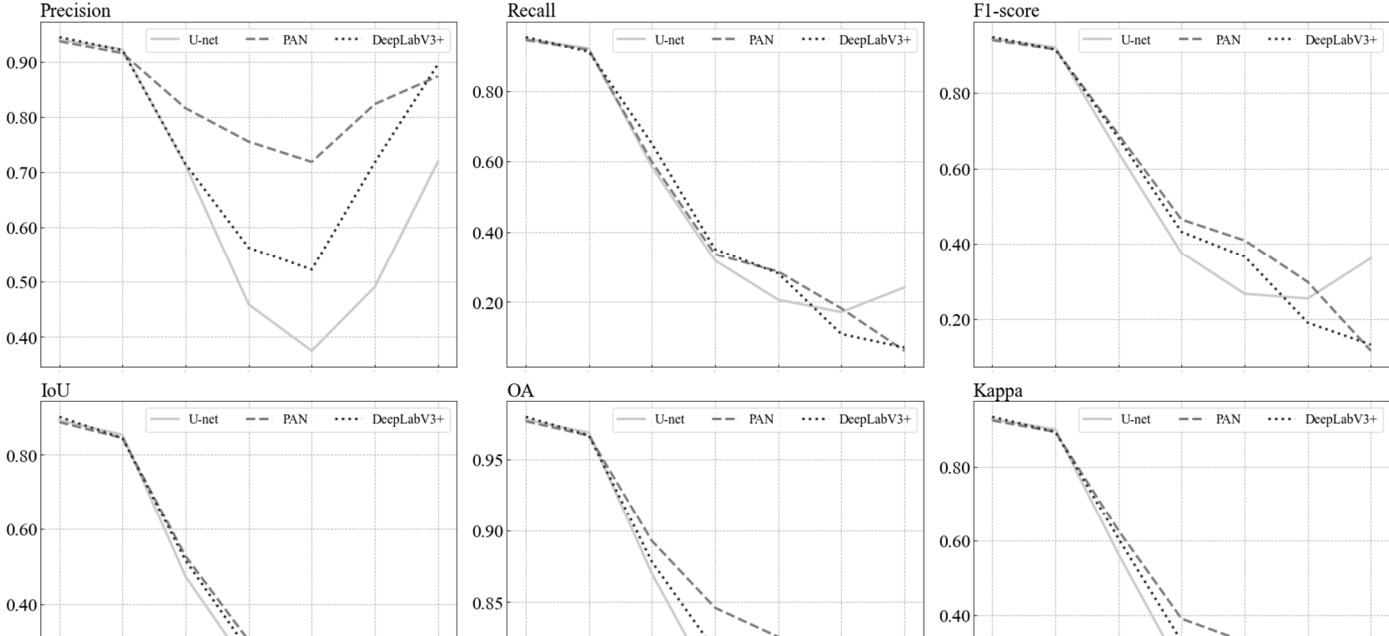

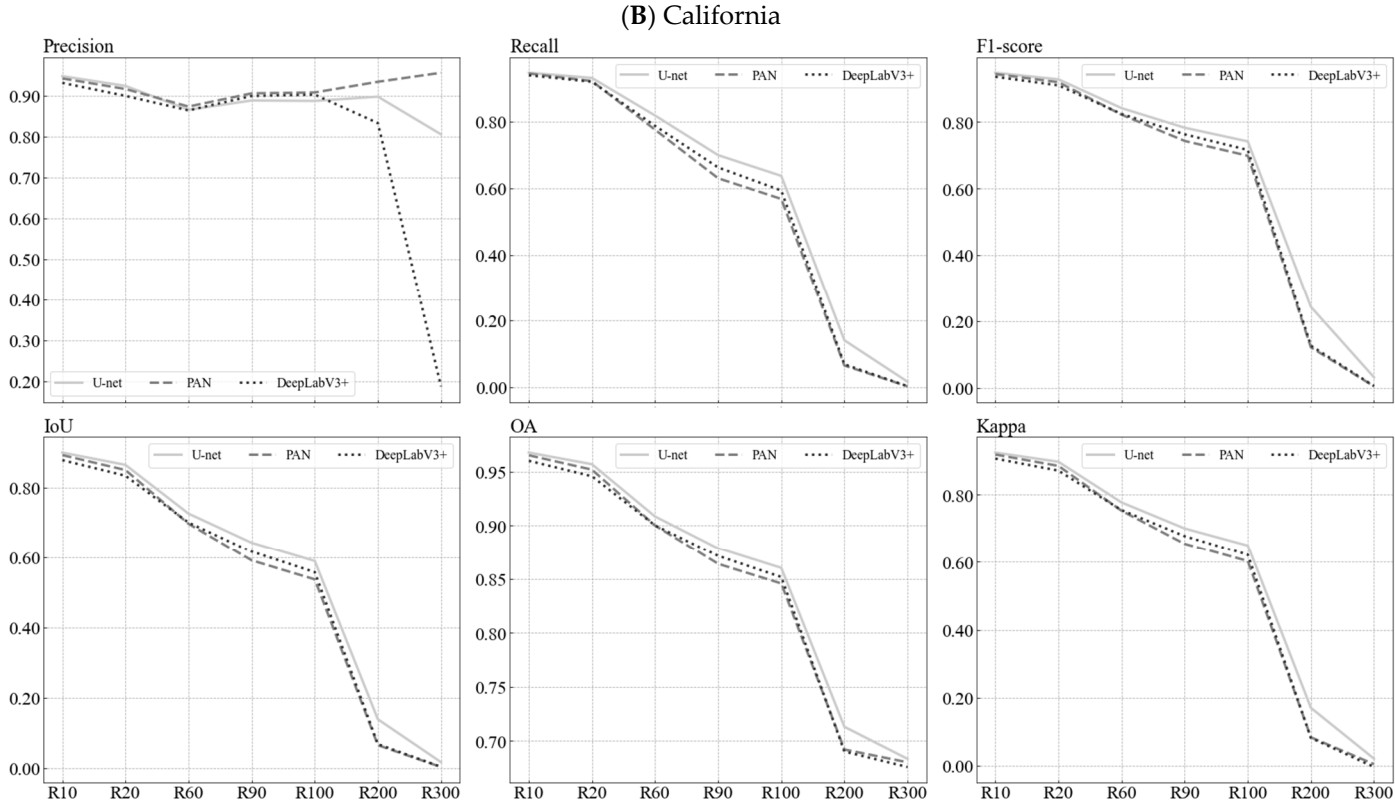

**Figure 10.** The quantitative evaluation of distinguishing between irrigation and non-irrigation at other spatial resolutions based on the trained models using data of 10 m resolution. The spatial resolution changed from 10 to 300 m for (**A**) Washington and (**B**) California. The indices include Precision, Recall, F1-score, IoU, OA, and Kappa.

## 5. Discussion

Central pivot systems are commonly utilized in agricultural irrigation practices across the United States. In the past, many studies have focused more on fields served by central pivot and less on fields served by other types of irrigation, which may be because of its obvious characteristics and because it is relatively easily identified [16–19,53,62]. This study demonstrates that deep learning techniques could efficiently extract irrigated croplands, regardless of the type of irrigation being utilized.

In conjunction with this study, it is hypothesized that spectral features exert the most significant impact in terms of differentiating irrigated land from other land types, with texture features following suit, while the influence of morphological features was found to be minimal. In areas where ground objects were correctly classified, the color and tone of irrigated land and other land types differed distinctly, and the uniformity and directionality also varied. Conversely, in regions where classification errors were present, there was similarity in the color and tone and a resemblance in the distribution density and magnitude of change. The morphological features of irrigated land had their characteristics in different locations, but were often similar to other land types, especially in areas with many land types and low spatial resolution. The distribution patterns of different land classes could affect the differences in spectral and textural features, affecting the identification process of irrigated farmland. Further verification is needed to confirm these findings.

DeepLabV3+ demonstrated a superior performance in California compared to Washington. PAN and U-net exhibited relatively poorer Precision, Recall, and F1-score metrics in California than in Washington. Conversely, the OA and Kappa values in Washington were superior to those in California, as shown in Table 1. Given that the area of Washington is smaller and the sample size was comparatively smaller than that in California,

further research is necessary to assess how the intricacy of the regional environment and the sample size affect the deep models' abilities to learn and identify features of irrigated farmland.

We found that more complex and diverse environmental conditions in the coverage area and more training data could allow the deep models to perform well with temporal portability to extract irrigation information. The extraction effect of the deep models in California was better than that in Washington, except for May, when the extraction effect of U-net and DeepLabV3+ in Washington was better than in California. In different months, the metrics of the models in California fluctuated more slightly than those in Washington (S5 in Figure 7).

It has been shown that there is low overall accuracy in years with high rainfall during the crop-growing season (irrigation period), but very high accuracy in years in which dry summers occur [22]. Comparatively, this study shows that deep models could effectively identify irrigation when precipitation is high. The irrigation extraction using FCNs in wet conditions is needed in order to study this further.

In this study, we extracted irrigated croplands from high-resolution images. Even with high-resolution images, precise interpretation of the true nature of ground cover may not always be possible, potentially impacting our study. There have been several studies using high-resolution images and deep learning methods to extract irrigation [52,53,59]. These studies have achieved an overall accuracy exceeding 85%. Consequently, we project that errors introduced by our methodology should fall within acceptable parameters.

## 6. Conclusions

This study explored the FCNs + DCNNs methods for mapping irrigated areas from Sentinel-2 images. We analyzed the extraction results of the fields with varying distribution states and the fields served by different types of irrigation. Furthermore, we investigated the portability of deep models in remotely sensed images of different years and spatial resolutions.

We designed the models based on the popular FCNs architectures, i.e., U-net, PAN, and DeepLabv3+. In order to adapt to the irrigation detection task, we employed a deep enough ResNet-101 as the backbone. In the backbone, we discarded some pooling layers and adopted dilation convolution to preserve as much location information as possible. Bilinear interpolation operation was used for the up-sampling layers, and pre-training weights from the ImageNet dataset helped to expedite model training. The results show that irrigation could be effectively identified regardless of whether or not crops were growing in the irrigated fields. The deep models were able to distinguish between irrigation and other land classes, particularly rainfed farmland. On the other hand, similar spectral, textural, and morphological features, as well as discrete and heterogeneous distribution, still interfered with the discrimination ability of the deep models. The deep models were able to effectively extract the irrigated fields served by most types of irrigation. The quantitative metrics were high, with IoU ≥ 0.88 and Kappa ≥ 0.91.

The study shows that the deep models' temporal portability became negatively affected as the time interval between the testing acquisition and training data increased. The lowest values of Recall and OA between 2018 and 2021 were 0.91 and 0.87, and the lowest values of the other metrics were 0.73. This study found that the performance of the deep models decreased as the difference in spatial resolution between the validation and training data increased. The metrics showed little variation in two states, where the resolution varied from 10 to 20 m. At resolutions ranging from 10 to 60 m, the lowest value of OA was 0.87. In Washington, at resolutions from 10 to 300 m, the lowest value of OA was 0.76. The comparison of different models suggested that the spatial resolution portability of the deep models could be improved by designing the model architecture. This study shows the potential of FCNs + DCNNs methods for mapping irrigated croplands across large areas and provides a reference for mapping irrigation.

The models were not specifically designed to meet the needs of the study areas, meaning that their application in other regions might face challenges due to differences in climate, soil types, crop types, irrigation methods, and even the quality of the available remote sensing data. Therefore, before applying the models to other regions, it would be advisable to conduct a validation process to assess their performances in those specific contexts.

**Author Contributions:** Conceptualization, W.L. and Q.X.; methodology, W.L. and Y.S.; software, W.L.; validation, W.L. and Y.Z.; writing—original draft preparation, W.L.; writing—review and editing, Y.S., Q.X. and L.G.; visualization, W.L.; funding acquisition, Y.L. All authors have read and agreed to the published version of the manuscript.

**Funding:** This research was supported by the Strategic Priority Research Program of the Chinese Academy of Sciences (XDA20020101) and the Third Xinjiang Scientific Expedition Program (2021xjkk0603).

**Data Availability Statement:** Sentinel-2 data as input images of the models are available from Google Earth Engine (GEE) (accessed in 2020 for Washington and in 2019 for California). Irrigated cropland layer data as labels of the models are available at https://agr.wa.gov/departments/land-and-water/natural-resources/agricultural-land-use (accessed on 5 April 2023) (the state of Washington) and https://data.cnra.ca.gov/dataset/statewide-crop-mapping (accessed on 3 April 2023) (the state of California). Daymet V4 (Daily Surface Weather Data on a 1-km Grid for North America, Version 4) precipitation data are available from Google Earth Engine (GEE).

**Acknowledgments:** We sincerely thank the agencies and researchers that provide available spatial data of the irrigated farmland.

**Conflicts of Interest:** The authors declare no conflict of interest.

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
