# Peer review of "Mapping Irrigated Croplands from Sentinel-2 Images Using Deep Convolutional Neural Networks"

_remotesensing, doi:10.3390/rs15164071_

Round 1

Reviewer 1 Report (Previous Reviewer 1)

line 32: explain OA- Overall accuracy

lines 66-68: give examples of regions where your findings would be of interest

In Conclusions, clarify if your findings have limits on extrapolating its application to other regions, given that the procedure was not created to meet the needs of the investigated areas, the states of Washinton and California.

Author Response

Reviewer 2 Report (New Reviewer)

This study uses DCNNs to extract irrigated cropland from Sentinel-2 imagery of Washington and California states in the United States.

 The topic is quite original and relevant in the field.

Similar approaches have existed in the specialized literature, but the complexity and number of comparative case studies make the paper relevant.

The conclusions consistent with the evidence and arguments
presented and do they address the main question posed .

Author Response

Reviewer 3 Report (New Reviewer)

English is very very hard to understand. The paper is full of gramma errors. I feel difficult to learn the idea and details of the manuscript. I do not think it is a wise decision to publish the paper.

 The edit trace does still exist in the manuscript, which make it even harder to follow.

In the left panel of Fig.1, the places of CA and WA are wrong.

Fig. 1 CA and Wa is wrong.

In my opinion, FCN is a type of DCNN, and U-Net is different from FCN. The FCN+DCNN architecture makes me confused.

The contribution of the paper seems unclear and insignificant.

English is very very hard to understand. The paper is full of gramma errors. I feel difficult to learn the idea and details of the manuscript. I do not think it is a wise decision to publish the paper.

Author Response

Reviewer 4 Report (New Reviewer)

I saw that the authors have made some improvements to the manuscript. However, I cannot recommend this work for publication at this time because the manuscript is very sketchy in its description of the field work. The authors need to address the following concerns I have:

1. Figure 1: How was the Irrigation distribution map obtained? Is it from land cover products? What is the size of the study area?

2. Did the authors conduct field survey for irrigated croplands and other land types? What types of land cover did the authors investigate? What are the survey methods? Where, when, and how many site are the investigations conducted? What are the typical interpreted signs of the feature types? The authors also need to show a map of the spatial distribution of the sample sites/regions surveyed.

3. I see that the authors extracted irrigated croplands and other types from the high resolution images. Even for high resolution images, it may not be possible to accurately interpret the true ground cover type, how can the authors ensure the accuracy? Also, what is the amount of training data obtained based on this method?

4. Similarly, how many sample data were acquired by the authors in the statewide agricultural land geodatabase?

5. The authors obtained training data from multiple sources, but the authors did not specify how many data were used for training in building the classification model? And how many are used for validation?

6. The authors only show the classification results of typical sample regions, and do not map the irrigation distribution of the study area.

7. Line 48-64: The authors mentioned some land cover products. I suggested that the authors compare the classification results of this study with the above products in terms of classification accuracy and consistency.

Minor editing of English language required.

Round 2

Reviewer 3 Report (New Reviewer)

The English needs to be further edited.

The impacts of global climate change and urban expansion on cropland should to be discussed to highlight the significance and urgency of this work. Following literatures are suggested.

 Evaluating trends, profits, and risks of global cities in recent urban expansion for advancing sustainable development. Habitat International….

Future urban land expansion and implications for global croplands. P. Natl. Acad. Sci. USA

Extensive editing of English language required

Author Response

Reviewer 4 Report (New Reviewer)

I believe the authors have made some improvements for their manuscript. However, there are several major issues with this work that have not yet been addressed. First, this work lacked field surveys; second, they were unable to complete mapping within the whole region; third, the authors need to compare with other land use products to further convince the advantages and accuracy of the mapping algorithms.

Minor editing of English language required

Author Response

This manuscript is a resubmission of an earlier submission. The following is a list of the peer review reports and author responses from that submission.

Round 1

Reviewer 1 Report

The main question addressed by the study described in this manuscript is the feasibility of using current high spatial resolution satellite imagery to delineate irrigated croplands in function of the methods used for irrigation (center pivot, sprinkler, drip etc.).
Good solutions to this question are already published, the authors are aware and cite a part of them. In this context, the originality of the submitted study would consist in the use of advanced methods of deep learning (DL) with capitalization of semantic information, in addition to the spectral and spatial information.
The authors have chosen the area of study in the United States, where different irrigation methods are currently used, in order to train the proposed DL algorithms. In addition, accurate information is made available thanks to the Irrigation and Water Management Survey, for their verification.

However, in my opinion this study has several problems:

- I do not see application or value for the very area of study (because it already has very good data) and I do not find in the manuscript what would be other areas with different irrigation methods, but with limitation of field data for mapping. Please clarify what areas are you considering.

-   I find it confusing that you propose as a goal “practical applications" (e.g., water management - in this respect, you give citations, [20] etc.), although you only use optical data. So, you cannot monitor the areas of interest when there are clouds. It is confusing because, on the one hand, you propose TO MAP taking into account the irrigation methods, and, on the other hand, you also refer TO MONITOR the irrigated fields for practical applications, which you fail to do. Please rethink and clarify this aspect.

-   I have doubts as to how the data for verification (e.g., from the above Survey you cited as [59]) have been used, because some are obviously misunderstood, or at least miscited (e.g., lines 167-169, or lines 178-179). Please review the entire sequence of data processing and interpretation.

Reviewer 2 Report

In this study, the authors use DCNN models based on Sentinel-2 image data to extract irrigated land with different distribution states and different irrigation methods in Washington and California. The authors investigate the portability of deep models in remotely sensed images of different acquisition times and different spatial resolutions. The work is well organized, the methodology, research, and discussion are correctly documented and easy to read, and the work contains new information for the scientific community.

Author Response

We thank you for the comments on the manuscript. The manuscript has been updated.

Reviewer 3 Report

For me may be of interest to the community but in its current state it lacks rigor and details.

I recommend that the authors rework the article and submit a new version that takes into account the comments made in the attached pdf

Author Response

We thank you for the comments on the manuscript. The manuscript has been updated. Please see the attachment

Reviewer 4 Report

The article is well structured, clearly and explicitly written, with plenty of good illustrations. It looks like coming through solid proofreading and makes a good general impression. 

Author Response

(The authors gave the same response as above.)

Round 2

Reviewer 1 Report

In my opinion, the concept underlying this study on mapping irrigated croplands is not solid. It seems to me that this application on irrigation has served for developing new DL algorithms, which may be appropriate for other types of applications.

As I wrote previously, I do not see any interest of the proposed subject in the very areas of study, the states of Washington and California, because they already have comprehensive data of very good quality collected on the ground.

Meanwhile, the authors ‘reply did not clarify to me what other areas worldwide are targeted to apply their present findings.

…. The assessments of the temporal and spatial resolution portability are interesting, but what are the characteristics of the areas in need for detailed maps of irrigated crops to apply the results? In other terms, what are the constraints of the proposed DL model, which is not intended for Washington or California?

This is very difficult to understand from the text and that is why I wrote that the underlying concept is not solid.

For example,
the claim made at lines 303-309 is debatable because the thresholds and timing of the peek greenness are crop-specific.  In addition, there are more than one crop season in the two states investigated, aren't they? In fact, what is mapped? Cereals, Orchards, Vineyards, Fruits & Vegetables, etc.? I ask because the information is missing in the manuscript, and I didn’t find it because (1) for Washington, the link with information at line 219 is broken, and (2) for California, the similar link at line 236 give the message "Access denied".

Reviewer 3 Report

The article has been improved but additional clarifications are necessary.
